# Who is missed in a community-based survey: Assessment and implications of biases due to incomplete sampling frame in a community-based serosurvey, Choma and Ndola Districts, Zambia, 2022

Natalya Kostandova[1]*, Simon Mutembo[2], Christine Prosperi[2], Francis Dien Mwansa[3], Chola Nakazwe[4], Harriet Namukoko[5], Bertha Nachinga[4], Gershom Chongwe[6], Innocent Chilumba[7], Kalumbu H. Matakala[8], Gloria Musukwa[9], Mutinta Hamahuwa[10], Webster Mufwambi[11], Japhet Matoba[12], Kenny Situtu[6], Irene Mutale[6], Alex C. Kong[2], Edgar Simulundu[9], Phillimon Ndubani[9], Alvira Z. Hasan[2], Shaun A. Truelove[1,2], Amy K. Winter[2,13], Andrea C. Carcelen[2], Bryan Lau[1], William J. Moss[1,2], Amy Wesolowski[1]

1 Department of Epidemiology, Johns Hopkins University Bloomberg School of Public Health, Baltimore, Maryland, United States of America, 2 Department of International Health, International Vaccine Access Center, Johns Hopkins University Bloomberg School of Public Health, Baltimore, Maryland, United States of America, 3 Department of Immunizations, Ministry of Health, Government of the Republic of Zambia, Lusaka, Zambia, 4 Information, Research and Dissemination, Zambia Statistics Agency, Lusaka, Zambia, 5 Population and Social Statistics, Zambia Statistics Agency, Lusaka, Zambia, 6 Tropical Diseases Research Centre, Ndola, Zambia, 7 Biomedial Sciences Department, Tropical Diseases Research Centre, Ndola, Zambia, 8 Clinical Research Department, Macha Research Trust, Macha, Zambia, 9 Macha Research Trust, Macha, Zambia, 10 Clinical Research Laboratory Department, Macha Research Trust, Macha, Zambia, 11 Administration, Tropical Diseases Research Centre, Ndola, Zambia, 12 Molecular Biology Department, Macha Research Trust, Macha, Zambia, 13 Department of Epidemiology and Biostatistics, University of Georgia, Athens, Georgia, United States of America

* nkostan1@jh.edu

## Abstract

Community-based serological studies are increasingly relied upon to measure disease burden, identify population immunity gaps, and guide control and elimination strategies; however, there is little understanding of the potential for and impact of sampling biases on outcomes of interest. As part of efforts to quantify measles immunity gaps in Zambia, a community-based serological survey using stratified multi-stage cluster sampling approach was conducted in Ndola and Choma districts in May—June 2022, enrolling 1245 individuals. We carried out a follow-up study among individuals missed from the sampling frame of the serosurvey in July—August 2022, enrolling 672 individuals. We assessed the potential for and impact of biases in the community-based serosurvey by i) estimating differences in characteristics of households and individuals included and excluded (77% vs 23% of households) from the sampling frame of the serosurvey and ii) evaluating the magnitude these differences make on healthcare-seeking behavior, vaccination coverage, and measles seroprevalence. We found that missed households were 20% smaller and 25% less likely to have children. Missed individuals resided in less wealthy households, had different distributions of sex and occupation, and were more likely to seek care at health facilities. Despite these

**Data Availability Statement:** The individual survey data were collected under data sharing agreements from Zambia Ministry of Health and the Zambia National Health Research Authority. As per the Zambia Health Research Act, access to data requires approval from the Zambian National Health Regulatory Authority. To obtain this access, please contact Dr Victor Chalwe, Acting Director of the Zambia National Health Research Authority (victor.chalwe@nhra.org.zm).

**Funding:** This work was supported by the Bill and Melinda Gates Foundation (OPP1094816) (NK, SM, CP, FDM, CN, HN, BN, GC, IC, KHM, GM, MH, WM, JM, KS, IM, ACK, ES, PN, AZH, SAT, AKW, ACC, WJM), Burroughs Wellcome Fund (1015823.03) (AW), National Institute of Allergy and Infectious Diseases (1R01AI160780-01) (AW), the National Institutes of Health (DP2LM013102) (NK, AW), and the Department of Health and Human Services Public Health Service Ruth L. Kirschstein National Research Service Award (1T32AI165369-01A1) (NK). The funders had no role in study design, data collection, analysis, interpretation, writing of the manuscript, or decision to publish.

**Competing interests:** The authors have declared that no competing interests exist.

differences, simulating a survey in which missed households were included in the sampling frame resulted in less than a 5% estimated bias in these outcomes. Although community-based studies are upheld as the gold standard study design in assessing immunity gaps and underlying community health characteristics, these findings underscore the fact that sampling biases can impact the results of even well-conducted community-based surveys. Results from these studies should be interpreted in the context of the study methodology and challenges faced during implementation, which include shortcomings in establishing accurate and up-to-date sampling frames. Failure to account for these shortcomings may result in biased estimates and detrimental effects on decision-making.

## Introduction

Infectious disease transmission is dictated by the susceptibility of the population, pathogen transmissibility, and effective contact patterns. Key to pathogen control is understanding the landscape of population immunity and susceptibility. Serological surveys (serosurveys) provide the most direct measure of immunity and have been used to quantify susceptibility and understand the epidemiology of various pathogens, including SARS-CoV-2 [1], measles [2], pertussis [3], and tetanus [4,5]. Community-based serological surveys are often considered the gold standard study design to measure population-level outcomes more accurately [6,7].

Serosurveys are particularly relied upon for programmatic decision-making in the control of vaccine-preventable diseases, including the timing and targets of measles supplemental immunization activities [8]. Historically, measles control and elimination efforts were primarily implemented through routine immunization programs and supplemented by nationwide preventative Supplementary Immunization Activities (SIAs). In recent years, efforts have been undertaken to tailor these supplementary efforts, prioritizing specific subnational areas or population groups for intensive vaccination activities. However, identifying susceptible populations remains challenging, and no World Health Organization (WHO) regions has been able to maintain measles elimination despite previously stated ambitions to eliminate the disease [9–11]. Measles serology provides a more direct measure of population-level susceptibility to inform the risk of outbreaks and ultimately guide immunization strategies and policies [8]. However, the expense of community-based serosurveys, difficulty in execution, and high burden on human resources make them prohibitive to use as tools for routine surveillance, especially in settings with limited resources [6,7].

In Zambia, several measles serological studies using less resource-intensive convenience samples have been conducted to identify locations with high immunity gaps, despite overall high measles vaccine coverage nationally. These studies have used samples from laboratory-backed measles case surveillance [12], residual samples collected as part of the Zambia Population-Based HIV Impact Assessment (ZAMPHIA) [13], and convenience samples from hospital laboratories (study ongoing) to reveal immunity gaps across age and space. However, the extent to which these different convenience sampling approaches result in biased estimates of seroprevalence still needs to be determined, requiring careful study design and characterization [8]. Immunity estimates derived from convenience samples can diverge from those derived from population-based surveys due to differences in disease severity or exposure levels [14,15], participation rates [16,17], location of residence [18], and sociodemographic characteristics [19]. On the contrary, some studies have found little variation in estimates between convenience and population-representative samples [20,21], or that the amount of bias

introduced due to sampling bias can vary by outcome of interest [22,23]. However, even for community-based surveys, some individuals may be missed, which may lead to biases and limit the possibility of evaluating the validity of convenience sampling approaches [6,13,24–28]. Concerted efforts are needed to confirm the representativeness of community-based serosurveys to be able to validate both sampling approaches.

To understand the extent to which sampling bias may be present and affect the results of a community-based serosurvey conducted in two districts of Zambia in 2022, we conducted a follow-up study of households missed from the serosurvey's sampling frame. Specifically, we wanted to understand whether the two populations (those included in the original serosurvey and those missed from its sampling frame) differed along some characteristics. Further, we wanted to evaluate whether the exclusion of these households was likely to result in bias in outcomes of interest. Specifically, we compared the socio-demographic status, healthcare-seeking, and vaccination history of missed households and individuals to those included in the serosurvey. We further investigated if estimates of healthcare seeking for all age groups and measles seroprevalence for young children in the age range targeted by immunization campaigns would be biased by missing specific populations from the survey. Finally, we investigated the impact of reducing missingness in the sampling frame on the magnitude of imbalances present in key characteristics of individuals.

## Materials and methods

A graphical summary of study methods is provided in S1 Appendix.

### Original and follow-up (missed populations) studies

**Originally enrolled households.** A population-based serosurvey evaluating measles seroprevalence by age group was conducted in two districts in Zambia (Ndola and Choma) from April to June 2022. Choma District is primarily a rural district located in Southern Province. Ndola District is classified as an urban district located in Copperbelt Province, with the third highest population density in the country [29]. In each district, 36 clusters were selected using probability proportional to population size. To establish a household sampling frame, in each of the selected clusters, the data collection team conducted a two-day listing, during which data collectors went door-to-door to every household, and collected information on the household size and presence of individuals in different age groups and their willingness and ability to participate. If no adult member of the household was available during listing, up to two attempts to revisit this household were made during the two-day period. In some cases, when an adult was not available, information about the household was obtained from others that knew the household composition (e.g., neighbors). Within the listing stage each household fell into one of the four groupings: 1) Household was unable to be listed due to it being locked, or having no adult present and no other adult individual able to provide information about household make-up ("No contact"); 2) Household was listed, provided consent to participate in the study if selected, and stated that household members would be available during the study period; 3) Household that refused to be listed or to participate in the study if selected; or 4) Household that agreed to be listed but stated they would not be available during the study implementation time (e.g., due to upcoming travel). To ensure sufficient sample size in each age group, three sampling frames were established: households with adults 15 years or older (all listed households), those with children 1–4 years old, and those with children 5–14 years old. These sampling frames were restricted to households that provided consent and were available to participate in the study if selected. From each sampling frame, a pre-determined number of households was independently selected for each age group of interest. For a selected

household within a specific sampling frame, an eligible individual was randomly selected for that age group to be enrolled. A structured questionnaire was administered to the selected individual or to the caretaker of children, and dried blood spot samples were collected and tested using a commercial enzyme-linked immunoassay (Measles IgG Test System, Zeus Scientific, USA, 9Z9271GB). The data collection tool used included, to the extent possible, questions and wording from other nationwide questionnaires, including questions on healthcare seeking, history of fever and rash, as well as demographics, which were aligned with the 2018 Zambia Demographics and Health Survey [30] or the 2015 Living Conditions Monitoring Survey VII [31]. Other questions were either adapted or developed for the purpose of the survey, such as questions on healthcare seeking at district-level hospitals. The questionnaire was piloted during training. For households that were not listed due to no contact during listing, data collection team recorded the Global Positioning System (GPS) coordinates, description of location of the household, and name of the head of the household, when possible.

**Identification of missed households.**   Missed households were defined as those eligible for the study but not listed at the time of enrollment (e.g., the inhabitants were absent during listing) or those listed but who reported being unavailable during data collection (see Fig 1A). These households were excluded from the sampling frames since they had a zero probability of being selected to participate in the study. However, listed households that refused participation (15 households, <1%) were not eligible for the missing household study. In total, 2,738 households (23%) were classified as missed households.

For households with young children (1–4 years), we calculated that 288 households were needed for the missed population survey to detect at least a 10% difference in vaccination coverage for 1–4 year-olds with 80% power and alpha = 0.1, assuming vaccination coverage of 90% in the original study. We anticipated that reasons for missingness might vary between clusters with heterogenous levels of missed households and we divided clusters into tertiles based on the total number of missed households before selection. In Ndola District, four clusters were randomly selected per tertile. Because the average number of missing households was lower in Choma District, we increased the number of clusters selected to reach the desired sample size by oversampling clusters from medium and high tertiles. We randomly selected two, eight, and six clusters from the low, medium, and high tertiles, respectively, for a total of 16 clusters. Within each selected cluster, all missed households were eligible if the number of missed households was less than 40; otherwise, 40 households were randomly selected. Restricting the number of sampled households per cluster to 40 allowed us to reach the desired sample size in both districts while spending on average a week in each cluster and maintaining a feasible workload per team. We also did not believe that enrolling more than 40 missed households per cluster would provide us with effectively new information about the missed population in that cluster.

The missed population study collection occurred between July and August 2022 (see S1 Fig), about three months after the beginning of the original survey. Data collection teams were provided GPS coordinates for the selected households, the name of the household head (when available), and descriptions of the household collected during listing in the original study. Data were collected by a subset of data collection personnel involved in the original study to reduce interviewer bias and increase comparability between the two studies, as well as to capitalize on the team's experience with the questionnaire. Furthermore, we anticipated that the teams' experience and familiarity with the clusters would be an advantage in locating the missed households, especially where descriptions of the households available from the original study were vague. Finally, the teams had a clear understanding of the difficulties of working in the clusters, including road conditions, terrain, and distance, which we judged to be an advantage for the missed populations study.

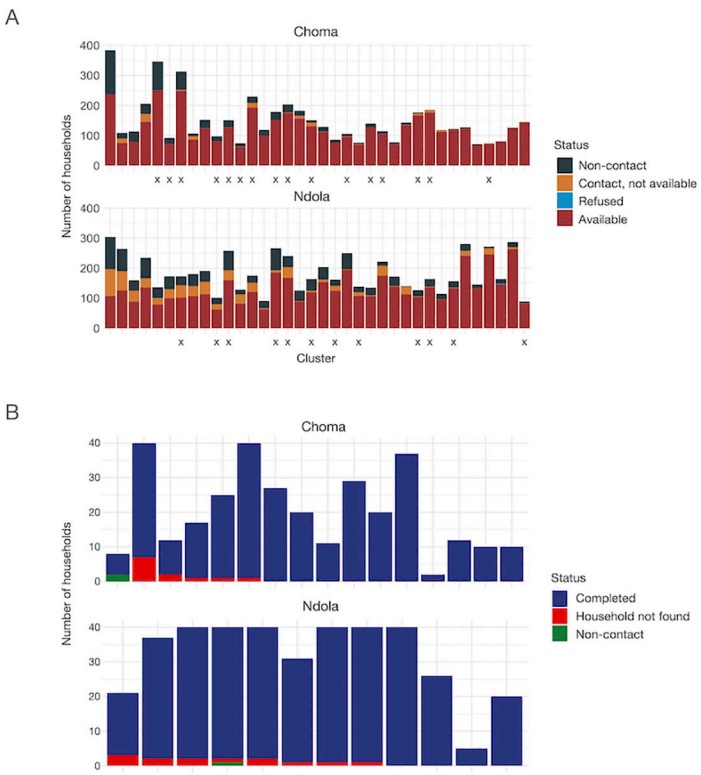

**Fig 1. Status of households enrolled in the original community-based measles serological survey and missed populations study, Ndola and Choma Districts, Zambia, 2022.** A. The distribution of household status from listing in the original serosurvey conducted in Choma and Ndola Districts, by cluster. Households classified as "Available" provided consent to participate in the study and reported that they would be available during the data collection; these households comprised the sampling frame for the original study. Households that refused ("Refused") were excluded from the original study sampling frame and were ineligible for the missed populations study. Households classified as "Non-contact" were households that were locked at the time of listing (and during revisits), or if there was no adult respondent at home, and nobody was available to provide information about the household (e.g. neighbor). Finally, households that were listed but which reported not being available during data collection ("Contact, not available") were excluded from the sampling frame in the original study. The households in the latter two categories were eligible for the missed populations study. Clusters are arranged in descending order by percentage of households eligible for the missed populations study ("Non-contact" and "Contact, not available" households). "X"'s indicate clusters selected for the missed population study. B. Distribution of households that the data collection team attempted to reach by status, cluster, and district in the missed populations study. Households classified as "Completed" were successfully located and provided consent to participate in the study. "Household not found" indicates households identified for inclusion in the missed populations study that could not be located during this study. "Non-contact" refers to households which were physically located, but ones in which the data collection team could not contact its occupants. No household refused participation. Clusters are arranged in order of decreasing percent missed in the missed populations study, comprised of "Household Not Found" and "Non-contact" households.

After locating the missed households, the teams enrolled one eligible individual from each age group of interest residing in the household (child 1–4 years old, child 5–14 years old, adults 15 years and older). If individuals agreed to participate, data collection teams conducted interviews using a shortened data collection tool from the original study, namely removing the module on travel survey and mobile phone ownership. No blood samples were collected, and no additional questions were added to this questionnaire.

## Operational definition of original and missed populations

Hereon forth, "original" population refers to the study sample obtained from the serosurvey carried out in April—June 2022. "Missed" population refers to the study sample included in the follow-up survey carried out in July—August 2022. These were households selected among those eligible for the original serosurvey but not listed or reporting to be unavailable during study period.

## Comparison between original and missed populations

We compared household and individual demographic characteristics and reported healthcare-seeking for all three age groups and vaccination history for children 1–4 years old. Definitions and categorizations of characteristics are provided in S2 Appendix. We calculated the proportion of respondents in each category or the mean and standard deviation across variables. Differences between populations were assessed using either Pearson's chi-squared test or Student's t-test. In addition, we fit a mixed effects model with the cluster as a random effect to test for associations between covariates and residing in a missed household. Details of the model are provided in S3 Appendix. For comparison of household-level characteristics, we restricted the original study to households that were selected from the sampling frame of adults (all households). All analyses were restricted to clusters selected for the missed populations study.

## Estimating bias from an incomplete sampling frame

To assess the impact of missed individuals and households from our sampling frame, we simulated surveys to include these populations in our original sampling frame. For each cluster, we selected with replacement the same number of individuals as the number of households selected in the original survey (see S1 Table). We bootstrapped to obtain 100 replicates using the original dataset ("Original" bootstrap) and 100 replicates using the mixed dataset ("Mixed" bootstrap), the latter comprised of both the original and missed population data sets. Households in the sampling frame were assigned different selection probabilities (S4 Appendix). After obtaining the "Original" and "Mixed" bootstraps, we calculated weighted and unweighted proportions for three outcomes of interest: proportion of individuals willing to seek care in one of the three referral-level hospitals in each study area (Arthur Davison Children's Hospital and Choma General Hospital for children 1–14 years old, and Ndola Teaching Hospital and Choma General Hospital for adults), stratified by age groups; coverage with second dose of measles-containing vaccine (MCV2) for children 1–4 years old; and measles seroprevalence for children 1–4 years old. In the original dataset, measles seropositivity was estimated from dried blood spot samples. Since no blood samples were collected as part of the missed populations study, seropositivity was predicted for children 1–4 years old using covariates such as individual and household demographics, healthcare seeking, vaccination history, and history of fever and rash from the original dataset (see S5 Appendix).

Bootstrap median and intervals for the difference in estimates between the "Original" and "Mixed" bootstrap simulations were obtained by first calculating the difference between estimates in each bootstrap replicate and then obtaining the 50th, 2.5th, and 97.5th percentiles. Bias was defined as the difference in the estimates obtained from the "Original" bootstrap and the "Mixed" bootstrap simulations.

## Implications of increasing completeness of the sampling frame

We explored a strategy to decrease potential bias from an incomplete sampling frame by simulating listing additional households until a defined missingness threshold was reached. For a

set threshold of missingness C%, if the original sampling frame for the cluster was missing less than C% households, listing was stopped in that cluster. However, if the original sampling frame was missing at least C% households in a cluster, listing continued until at most C% of households were excluded from that sampling frame.

We considered six thresholds of missingness (5%, 10%, 15%, 20%, 25%, and 30%). For each cluster, we calculated the additional number of households that would have to be listed from the missed populations dataset to reach each threshold of missingness considered. For each of the 100 simulations, we randomly selected that number of households from the missed households. We included them in the sampling frame of the households enrolled in the original study. We then bootstrapped using selection probabilities to sample the desired number of individuals from this updated sampling frame. We compared the estimate for the indicator of interest from the resulting bootstrap to the bootstrap obtained under the assumption that all households from the missed population study were also included in the sampling frame ("Mixed" bootstrap, as described above). We considered two indicators of interest, sex of adult respondents and the proportion of enrolled households with children, because these had the largest imbalance between the original and the missed populations study samples.

All data were collected using CSPro [32]. Data management, processing, and analyses were carried out in R version 4.2.1 [33] and RStudio version 2023.6.2.561 [34].

**Ethics statement.** The original study received human subjects research approval from Johns Hopkins Bloomberg School of Public Health IRB (IRB00018265). The missed populations study was also approved by this IRB (IRB00019834). Both studies were approved under a single protocol by the National Health Research Authority, Zambia, and the Tropical Diseases Research Centre Ethics Review Committee (IRB00002911, FWA0003729).

The recruitment period lasted from April 8, 2022 to June 2, 2022 for the original study, and from July 4, 2022 to August 4, 2022 for the missed populations study.

During household listing in the original study, verbal consent was obtained from the head of the household or another adult (18 years or older) present at the time of the listing to collect the household listing data. During the implementation of both the original and missed population studies, for adult participants, written informed consent was obtained. For individuals under 18 years old, parents or guardians provided parental permission for the child to participate. If the adult individual was unable to provide a signature, they provided a thumbprint to signify their consent. Verbal assent was obtained for children aged 10 to 17 years old, and was documented by the person obtaining assent. All consent, parental permission, and assent forms were available in English, Bemba, and Tonga, and were administered in the preferred language of the participants by trained study team members.

## Results

### Comparing socio-demographic characteristics and healthcare seeking between the original and missed populations

**There were differences in socio-demographic characteristics between residents of missed households and households participating in the original serosurvey.** In total, 1245 individuals were included in the analysis from the original serosurvey and 672 from the missed households study (see Table 1). Of the missed households selected for follow-up, most were enrolled (96% per district, Fig 1B); no households refused participation, but several (3.6%) were not found. Living in a female-headed household, lower wealth index, smaller household size, and living in Ndola (urban setting) rather than Choma District (predominantly rural district) were positively associated with being in the missed population (S2 and S3 Tables). For adults, being male was also significantly associated with residing in a missed household

**Table 1. Comparison of household-level characteristics of individuals enrolled in the original community-based measles serological survey and missed population study.** The original serosurvey was carried out in April—June 2022 in Ndola and Choma districts, Zambia, using stratified multi-stage cluster design. The follow-up missed population study was carried out in a subset of clusters of the original survey between July—August 2022. This study was carried out in a subsample of clusters from the original survey; in each selected cluster, a sample of households not available during listing of the original serosurvey, and hence excluded from its sampling frame, were randomly selected.

| Characteristic | Ndola | | | Choma | | |
|---|---|---|---|---|---|---|
| | Original, N = 454 | Missed Population, N = 366 | p-value[3] | Original, N = 791 | Missed Population, N = 306 | p-value[3] |
| Sex of head of household | | | 0.13 | | | **0.012** |
| Female | 33% | 38% | | 28% | 36% | |
| Male | 67% | 62% | | 72% | 64% | |
| Number of people in household[1] | 6 (2) | 4 (3) | **<0.001** | 6 (3) | 4 (3) | **<0.001** |
| Households with children | 92% | 60% | **<0.001** | 92% | 63% | **<0.001** |
| Number of children in household[1] | 2.23 (1.38) | 1.33 (1.49) | **<0.001** | 2.68 (1.79) | 1.60 (1.66) | **<0.001** |
| Wealth score[1,2] | 1.90 (1.16) | 1.32 (1.21) | **<0.001** | -1.31 (2.69) | -1.56 (2.79) | **<0.001** |

[1]Mean (SD).

[2]Wealth score: A standardized score calculated after principal component analysis of socio-economic variables.

[3]Calculated using Pearson's Chi-squared test; Wilcoxon rank sum test; Fisher's exact test.

(OR = 1.78; 95% CI: 1.34–2.36) (S3 Table). Model diagnostics show a moderately good fit, except for the model for children 5 to 14 years old (Hosmer-Lemeshow goodness-of-fit p-value <0.05) (S3 Appendix).

At individual level, children in the missed populations study came from less wealthy households (S2 and S3 Tables) but did not differ from children in the original survey on other socio-demographic characteristics. In contrast, adults differed in other characteristics besides wealth score (S4 Table). Notably, a smaller proportion of adult respondents in the missed populations study was female compared to the original study (54% vs. 71% in Ndola, p<0.001; 55% vs. 67% in Choma, p = 0.002). Adult respondents in the missed populations study also had a different distribution of occupations compared to the original study: a larger proportion of participants held elementary occupations, and a smaller proportion were retired, unemployed, or a homemaker (p = <0.001) (S4 Table).

**Individuals in missed households were more likely to seek care at referral-level hospital.** Since residual samples collected from hospital laboratories are commonly used, the original serosurvey focused on understanding the characteristics of individuals who might not seek care at these hospitals, leading to potential biases in estimating seroprevalence when using residual samples. As such, we further investigated differences in care-seeking behavior at these hospitals between participants in the original serosurvey and those residing in missed households. For all respondent groups, participants from missed households reported being more likely to seek care from the referral-level hospitals than participants in the original serosurvey (see Tables 2 and 3). Although government clinics and health centers were the predominant first choice for healthcare seeking in both the missed and original surveys, caregivers of children in missed households were more likely to first seek care for a child's febrile illness in these facilities than in the originally included households (Table 2). In Choma, adults from missed households were less likely to use public motorized transport and reported shorter travel times to health facilities (Table 3). While the studies were not powered to detect differences between adults stratified by whether they resided in households with or without children, stratification reduced the difference in characteristics of adults in missed and original studies (S5 Table).

**Table 2. Healthcare-seeking and characteristics reported by caregivers of children 1–4 years old and 5–14 years old.** Results presented are for univariable analysis, by district and age group, and multivariable analysis, by age group only. "Original" refers to the serosurvey carried out in April—June 2022 in Ndola and Choma districts, Zambia, using stratified multi-stage clustering design. "Missed" refers to the study sample from the follow-up missed population study, carried out in a subset of clusters of the original survey between July—August 2022. This study was carried out in a subsample of clusters from the original survey; in each selected cluster, a sample of households not available during listing of the original serosurvey, and hence excluded from its sampling frame, were randomly selected.

| Characteristic | Caregivers of children 1–4 years old | | | | | Caregivers of children 5–14 years old | | | | |
|---|---|---|---|---|---|---|---|---|---|---|
| | Ndola | | Choma | | Multivariable OR; p-value[2] | Ndola | | Choma | | Multivariable OR; p-value[2] |
| | Original (N = 101); Missed (N = 106) | p-value[1] | Original (N = 198); Missed (N = 105) | p-value[1] | | Original (N = 208); Missed (N = 185) | p-value[1] | Original (N = 371); Missed (N = 162) | p-value[1] | |
| *District (Ref: Ndola)* | | | | | **0.246; p = 0.001** | | | | | **0.104; p<0.001** |
| *Household-level characteristics* | | | | | | | | | | |
| Male-headed household | 73%; 67% | 0.32 | 82%;58% | **<0.001** | **0.513; p = 0.003** | 68%;32% | **0.027** | 70%;62% | 0.090 | **0.711; p = 0.030** |
| Wealth score[3] | 1.74 (1.26);1.35 (1.13) | **<0.001** | -1.36 (2.67);-2.00 (2.80) | **<0.001** | **0.821; p = 0.014** | 1.96 (1.12)–1.47 (1.09) | **<0.001** | -1.31 (2.74); -1.59 (2.91) | **0.003** | **0.651; p<0.001** |
| Number of people in household[3] | 6 (2); 6 (3) | 0.12 | **6 (3); 5 (2)** | **<0.001** | **0.919; p = 0.050** | 6 (3); 6 (2) | **0.016** | 6 (2); 5 (2) | **0.002** | **0.903; p = 0.002** |
| *Male child* | 41%;51% | 0.15 | 51%;58% | 0.24 | 1.458; p = 0.062 | 51%;46% | 0.30 | 48%;52% | 0.44 | 0.899; p = 0.477 |
| *Health care seeking at Choma General Hospital or Arthur Davison Children's Hospital* | | | | | | | | | | |
| Would seek care at referral-level health facility[4] | | **0.032** | | **0.004** | | | **0.032** | | **0.004** | |
| Don't know | 0%;0% | | 0.5%;0% | | | 0%;0% | | 0.3%;0% | | |
| No | 7%;1% | | 15%;3.8% | | | 5%;1% | | 14%;5.6% | | |
| Yes | 93%;99% | | 85%;96% | | | 95%;99% | | 85%;94% | | |
| Way to travel to referral-level health facility | | 0.43 | | 0.86 | | | 0.073 | | 0.20 | |
| Private motorized transport (e.g., personal car or scooter) | 5.0%;2.8% | | 2.5%;4.8% | | | 5.3%;8.1% | | 5.4%;8.0% | | |
| Public motorized transport (e.g., bus, taxi) | 85%;83% | | 72%;69% | | | 87%;82% | | 68%;60% | | |
| Walk | 8.9%;14% | | 21%;22% | | | 5.8%;9.7% | | 22%;28% | | |
| Bicycle | 0%;0% | | 3.5%;3.8% | | | 0%;0% | | 3.5%;2.5% | | |
| Other | 1.0%;0% | | 1.0%;1.0% | | | 1.9%;0% | | 0.5%;1.2% | | |
| Travel time to referral-level health facility[5] | | 0.23 | | 0.26 | | | 0.32 | | 0.13 | |
| Less than 30 mins | 17%;13% | | 17%;25% | | | 16%;18% | | 18%;19% | | |
| 30 mins—less than 1 hr | 35%;25% | | 18%;18% | | | 36%;29% | | 18%;26% | | |
| 1 hr—less than 2 hrs | 32%;39% | | 22%;26% | | | 33%;33% | | 22%;22% | | |
| 2 hrs—less than 3 hrs | 15%;23% | | 17%;13% | | | 15%;20% | | 22%;16% | | |
| 3 or more hours | 0%;0% | | 26%;17% | | | 0%;0% | | 21%;17% | | |
| *General healthcare-seeking behavior* | | | | | | | | | | |
| Money not a big barrier for health care seeking | 37%;47% | 0.12 | 24%;10% | **0.003** | 0.907; p = 0.684 | 33%;42% | **0.045** | 23%;10% | **<0.001** | 0.995; p = 0.980 |

*(Continued)*

**Table 2.** (Continued)

| Characteristic | Caregivers of children 1–4 years old | | | | | Caregivers of children 5–14 years old | | | | |
| | Ndola | | Choma | | Multivariable OR; p-value[2] | Ndola | | Choma | | Multivariable OR; p-value[2] |
| | Original (N = 101); Missed (N = 106) | p-value[1] | Original (N = 198); Missed (N = 105) | p-value[1] | | Original (N = 208); Missed (N = 185) | p-value[1] | Original (N = 371); Missed (N = 162) | p-value[1] | |
|---|---|---|---|---|---|---|---|---|---|---|
| Distance not a big problem for health care seeking | 71%;78% | 0.24 | 42%;51% | 0.11 | | 75%;83% | 0.059 | 44%;47% | 0.56 | |
| Where respondent would seek care for child's severe fever first | | 0.93 | | **0.041** | | | **0.021** | | **0.042** | |
| Government clinic / health center | 94%;95% | | 92%;98% | | | 92%;96% | | 94%;96% | | |
| Government or mission hospital | 4.0%;3.8% | | 3.0%;0% | | | 7.2%;2.2% | | 1.6%;1.9% | | |
| Pharmacy or chemist | 1.0%;0% | | 0%;0% | | | 0%;0% | | 0%;0% | | |
| Private clinic or hospital | 1.0%;0.9% | | 0.5%;0% | | | 0.5%;1.6% | | 0%;0% | | |
| Community health worker | 0%;0% | | 4.5%;1.0% | | | 0%;0% | | 4.0%;0.6% | | |
| Traditional healer | 0%;0% | | 0%;1.0% | | | 0%;0% | | 0%;0.6% | | |
| Other facility | 0%;0% | | 0%;0% | | | 0%;0.5% | | 0%;0.6% | | |
| Don't know | 0%;0% | | 0%;0% | | | 0%;0% | | 0.3%;0% | | |
| Has vaccination card[6] | | 0.82 | | 0.084 | | | | | | |
| No | 3.0%;1.9% | | 5.6%;3.8% | | | | | | | |
| Yes, card not seen | 36%;39% | | 33%;46% | | | *Only applicable to children 1–4 years old. Caretakers of children 5–14 years old were not asked these questions.* | | | | |
| Yes, card seen | 61%;59% | | 62%;50% | | | | | | | |
| Regular vaccination location[5,6] | | **0.004** | | 0.74 | | | | | | |
| Community health worker | 2.0%;0% | | 0%;0% | | | | | | | |
| Government clinic / health post | 80%;94% | | 86%;83% | | | | | | | |
| Government or mission hospital | 4.0%;0.9% | | 2.5%;1.9% | | | | | | | |
| Other facility or individual, specify | 0%;0.9% | | 0%;0% | | | | | | | |
| Outreach posts | 14%;3.8% | | 11%;15% | | | | | | | |
| Does not receive vaccines | 0%;0% | | 0.5%;0% | | | | | | | |

[1]Fisher's exact test; Pearson's Chi-squared test.

[2]In multivariable model, the following covariates were included: District, male sex, male-headed household, wealth score, number of people in household, money not a big problem for seeking healthcare. For children 1–4 years old only, receipt of second dose of Pentavalent vaccine was also included. Full results are presented in S3 Appendix.

[3]Mean (SD).

[4]Choma General Hospital for respondents in Choma District or Arthur Davison Children's Hospital for respondents in Ndola District.

[5]Excludes "Don't know" responses.

[6]Only applicable to children 1–4 years old. Caretakers of children 5–14 years old were not asked these questions.

**Table 3. Healthcare-seeking and characteristics reported by adults 15 years and older.** "Original" refers to the serosurvey carried out in April—June 2022 in Ndola and Choma districts, Zambia, using stratified multi-stage clustering design. "Missed" refers to the study sample from the follow-up missed population study, carried out in a subset of clusters of the original survey between July—August 2022. This study was carried out in a subsample of clusters from the original survey; in each selected cluster, a sample of households not available during listing of the original serosurvey, and hence excluded from its sampling frame, were randomly selected.

| Characteristic | Ndola | | | Choma | | | Multivariable OR; p-value[3] |
|---|---|---|---|---|---|---|---|
| | Original, N = 186[1] | Missed, N = 367[1] | p-value[2] | Original, N = 347[1] | Missed, N = 305[1] | p-value[2] | |
| *District (Ref: Ndola)* | | | | | | | **0.073; p<0.001** |
| *Household-level characteristics* | | | | | | | |
| Male-headed household | 66% | 62% | 0.39 | 71% | 64% | 0.057 | **0.737; p = 0.039** |
| Wealth score[4] | 2.00 (1.07) | 1.32 (1.21) | <0.001 | -1.36 (2.72) | -1.57 (2.80) | **0.005** | **0.602; p<0.001** |
| Number of people in household[4] | 5 (2) | 4 (3) | <0.001 | 5 (3) | 4 (3) | <0.001 | **0.858; p<0.001** |
| *Male respondent* | 29% | 46% | <0.001 | 83% | 64% | <0.001 | **1.776; p<0.001** |
| *Health care seeking at Choma General Hospital or Ndola Teaching Hospital* | | | | | | | |
| Would seek care at referral-level health facility [5] | | | **0.001** | | | <0.001 | |
| Don't know | 0.5% | 0% | | 0% | 0% | | |
| No | 6.5% | 1.4% | | 12% | 4% | | |
| Yes | 93% | 99% | | 88% | 96% | | |
| Way to travel to referral-level health facility | | | 0.29 | | | <0.001 | |
| Private motorized transport (e.g., personal car or scooter) | 6.5% | 6.5% | | 2.3% | 9.2% | | |
| Public motorized transport (e.g., bus, taxi) | 90% | 92% | | 72% | 59% | | |
| Walk | 2.7% | 1.9% | | 22% | 28% | | |
| Bicycle | | | | 3.5% | 3.6% | | |
| Other | 1.1% | 0% | | 0.9% | 0.3% | | |
| Travel time to referral-level health facility[6] | | | 0.53 | | | <0.001 | |
| Less than 30 mins | 12% | 13% | | 16% | 25% | | |
| 30 mins—less than 1 hr | 43% | 42% | | 14% | 22% | | |
| 1 hr—less than 2 hrs | 37% | 40% | | 24% | 23% | | |
| 2 hrs—less than 3 hrs | 7.9% | 4.4% | | 26% | 13% | | |
| 3 or more hours | 0.6% | 0.6% | | 19% | 14% | | |
| *General healthcare seeking* | | | | | | | |
| Money not a big barrier for health care seeking | 31% | 48% | <0.001 | 32% | 12% | **0.001** | 1.203; p = 0.265 |
| Distance not a big problem for health care seeking | 64% | 83% | <0.001 | 40% | 48% | 0.00 | |

[1]%.

[2]Fisher's exact test; Pearson's Chi-squared test.

[3]In multivariable model, the following covariates were included: District, male sex, male-headed household, wealth score, number of people in household, money not a big problem for seeking healthcare. Full results are presented in S3 Appendix.

[4]Mean (SD).

[5]Choma General Hospital for respondents in Choma District or Ndola Teaching Hospital for respondents in Ndola District.

[6]Excludes "Don't know" responses.

**Young children residing in missed households had similar vaccination coverage histories compared to children in the original study, although the site of vaccination differed.** The distribution of vaccine receipt for bacille Calmette-Guérin (BCG); the first dose of diphtheria, pertussis, and tetanus vaccine (Penta1); and first and second doses of measles-containing vaccine (MCV1, MCV2) was similar for children 1–4 years old in missed and original surveys (Fig 2). In Choma District, a smaller proportion of children from missed households had documented Pentavalent vaccine coverage compared to children in the original study

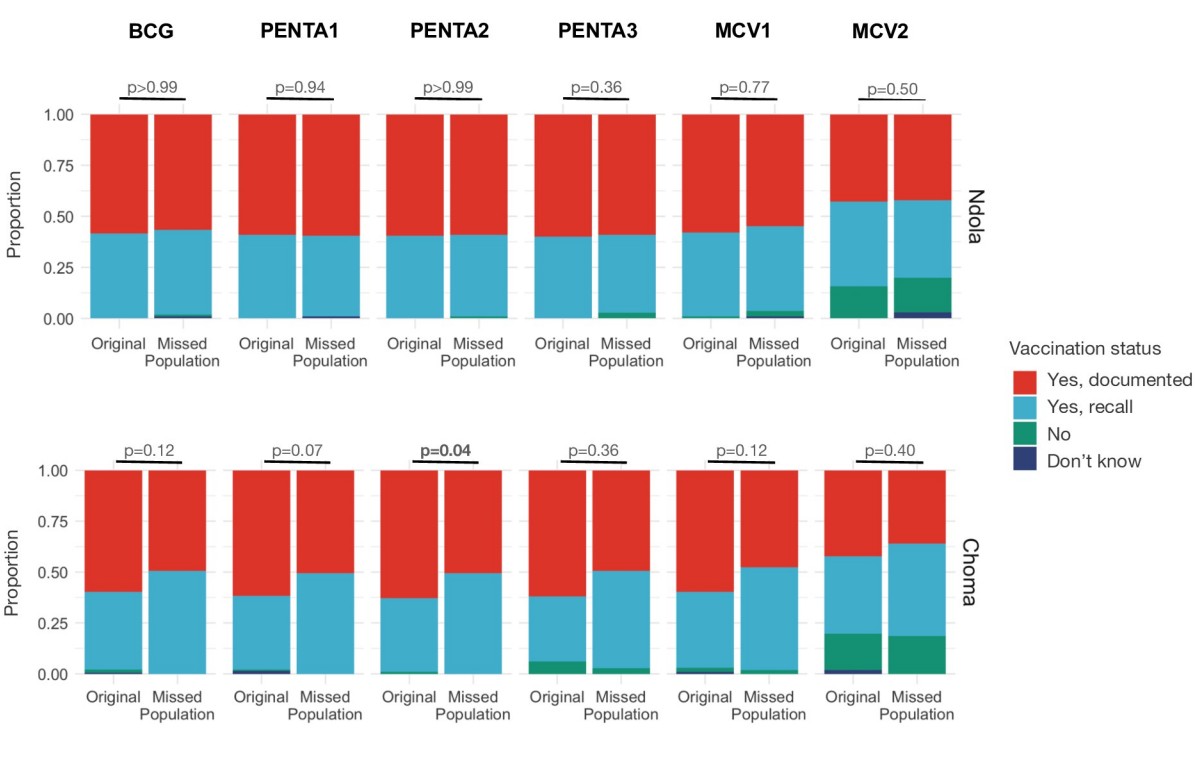

**Fig 2. Vaccination status for children 1–4 years old, by vaccine.** Results are presented for the original community-based measles serological survey and missed populations study, Ndola and Choma Districts, Zambia, 2022. The vaccines considered are bacille Calmette-Guérin, BCG; the first, second, and third dose of diphtheria, pertussis, and tetanus (Penta1, Penta2, Penta3); and first and second doses of measles-containing vaccine (MCV1, MCV2).

(Penta1: p = 0.073; Penta2: p = 0.044; Penta3: p = 0.021), although reported coverage was similar when sources of documentation (e.g., documented vs. recall only) were combined (Fig 2). In Ndola District, a larger proportion of caregivers in missed households brought children to government clinics or health posts as their regular vaccination location compared to those in the original study, and a lower proportion used outreach posts (p = 0.004) (Table 2).

## Estimating bias introduced by an incomplete sampling frame

Using bootstrapping to simulate the inclusion of missed households in the sampling frame along with those captured in the original study, we evaluated possible bias resulting from the incomplete sampling frame for three outcomes of interest: care seeking from specified referral-level facilities by each age group, MCV2 coverage for children 1–4 years old, and predicted measles seroprevalence for children 1–4 years old. Simulations that included missed households in the original study sampling frame resulted in a small change in estimates for all three outcomes (Fig 3), but the bias resulting from the exclusion of missed households from the sampling frame was minimal (<5% for all outcomes) and not statistically significant (S7 Table).

## Implications of increasing completeness of the sampling frame

We assessed the effect of the missingness threshold on bias in estimates of two characteristics: sex of adult respondents and the proportion of enrolled households with children. These characteristics were selected because they had a large imbalance between the missed households

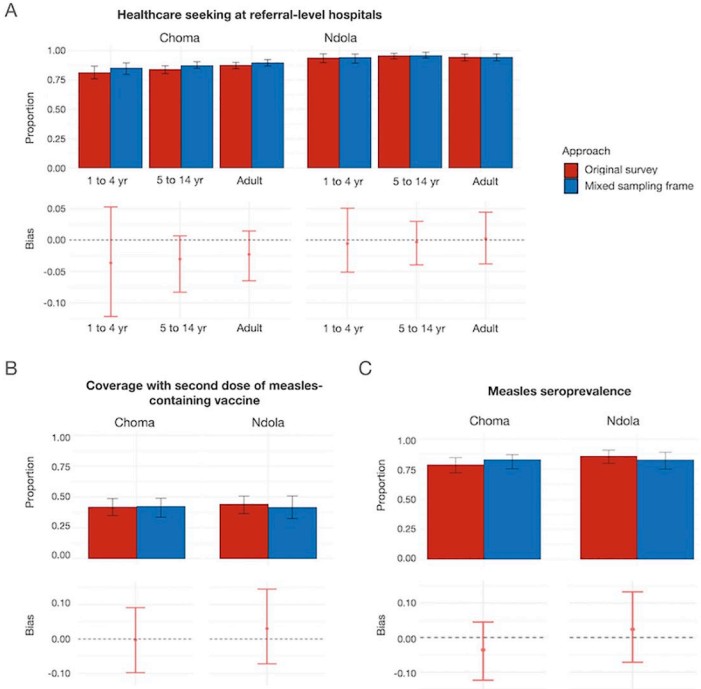

**Fig 3. Estimates of outcomes of interest using the sampling frame from the original community-based measles serological survey (excluding missed households) in Ndola and Choma Districts, Zambia, 2022, and a mixed sampling frame (including both households enrolled in the missed population study and households enrolled in the original study).** Weighting was done using the estimated population in each age group in each cluster in the missed population study for outcomes of interest including A. Healthcare seeking (actual and theoretical) at facilities of interest (Arthur Davison Children's Hospital and Choma General Hospital for children 1–4 and 5–14 years old, and Ndola Teaching Hospital and Choma General Hospital for adults 15 years and older), B. MCV2 coverage, children 1–4 years old, and C. Measles seroprevalence, children 1–4 years old.

and those in the original study (S4 Table). For both indicators, bootstrapped confidence intervals were wide and simulation results suggested that even the original study design (excluding all missed households) produced estimates that were not significantly biased (Fig 4). As expected, using smaller thresholds of missingness resulted in lower median bias (Fig 4). In Ndola District, using the original study design would have resulted in a median bias of just over 5% when estimating the proportion of adult respondents 15 years of age and older who were female, but using thresholds of missingness of 20% and lower would have brought the median bias within 5% of the value estimated if all missed households were included in the sampling frame (Fig 4A). In Choma District, the original study design would have resulted in a median bias of just over 2.5% when estimating the proportion of adults living in households with children. Using a 30% or lower threshold would have brought this median bias to below 2.5% (Fig 4B).

## Discussion

While community-based surveys with probability-based sampling are usually held as gold standard study design for estimating population-level outcomes, their validity may be threatened by incomplete sampling frames. In our study, the households and individuals missed in the original serosurvey had significantly different characteristics compared to those included in the original serosurvey. Households missing from the original serosurvey's sampling frame

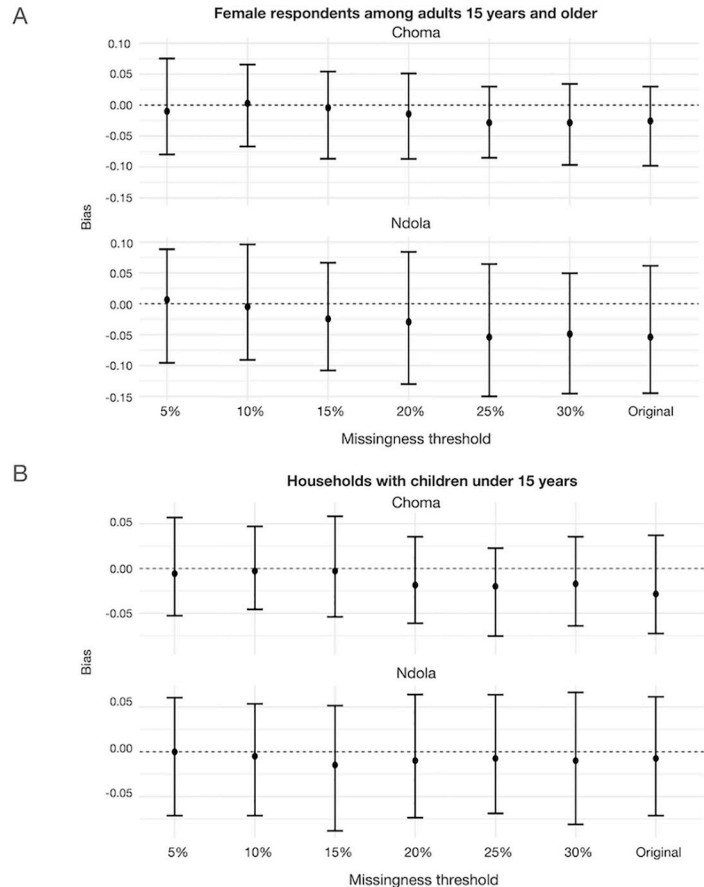

**Fig 4. Amount of bias for estimates of outcome of interest using different missingness thresholds to trigger additional listing, by district.** Outcomes of interest are A. female respondents among adults 15 years and over, and B. households with children under 15 years of age. Bias is defined as the difference in the indicated estimate in the original dataset (community-based measles serological survey carried out in Ndola and Choma Districts, Zambia, 2022), or bootstrapped with a specified percent missingness threshold and the estimate from a dataset with a sampling frame that includes all missed population households.

were smaller, less likely to have children, and less wealthy. Adults included in the missed population study were more likely to be male than in the original study and were more likely to hold elementary occupations. Because the original serosurvey had a secondary objective of identifying the profile of individuals who may not seek care at reference facilities, we compared enrollees' health-seeking attitudes and behavior in the original study and the missed population study. Residents of missed households were more likely to seek care at one of the referral-level hospitals of interest.

While residents of the missed households and those in the original study were different across these dimensions, simulations of what would have happened if missed households were included in the sampling frame indicate that critical outcomes of interest, such as healthcare-seeking behaviors across all age groups and routine MCV2 coverage and measles seropositivity among children 1–4 years old would not have changed significantly. This is likely because the number of households and individuals excluded from the sampling frame was relatively low overall, although this varied by cluster. Furthermore, because households missed generally had fewer residents and fewer children, their contribution to child-specific indicators was relatively low.

This provides evidence that the results of the serosurvey conducted in Ndola and Choma Districts are likely robust to these biases. However, observed imbalances in population characteristics, such as the sex of the respondents and the proportion of households with children, may contribute to biases in other studies where outcomes of interest are strongly associated with these characteristics. For outcomes like measures of population immunity, even a small bias could result in misclassification of subnational areas' risk for an outbreak if the outcome is close to a defined threshold of acceptability. We proposed a practical way to assess the magnitude of missingness that should trigger additional listing to reduce potential biases in indicators of interest. The suggested framework can be used to determine the threshold at which listing should continue if there are concerns about imbalances between listed and unlisted households. For example, in conducting a study where the sex of adult respondents was closely associated with the outcome of interest, the household listing should continue until the percentage of unlisted households was less than 20% in Choma District and 15% in Ndola District, which would assure that the median bias in the proportion of female respondents was less than 2.5%. In practice, estimating the exact value of the threshold to use would require having some prior understanding of expected strength of association between characteristics of households and outcome of interest. If this information is available, and data on these characteristics are collected during listing, simulations similar to the ones carried out in this study could be used to determine the threshold of missingness.

While we did not formally evaluate reasons why households included in the missed populations study were missing from the original study, data collection teams reported that a common reason for initially being missed was leaving households very early (5 or 6 in the morning) and returning home late, especially when household members were businessmen, farmers, and gardeners who tended to fields and plots outside of their households or traveled to markets or out of town to sell wares. At other times, residents of missed households were mobile due to their occupations: soldiers, bus and truck drivers, bus conductors, teachers on leave for examinations or vacation, or temporary workers. Understanding the demographic characteristics of populations likely missed during listing could inform strategies to include these populations. For example, being flexible with the hours of data collection (conducting the survey early or very late, scheduling weekend appointments) and location of data collection (at the workplace rather than at the household) might allow data collection teams to have higher response rates and fewer imbalances. Shifting between clusters to allow callback times allowed for more flexibility in reaching some households; however, this would be challenging at the listing stage.

The study has several limitations. In assessing bias, we only considered households and individuals missed due to unavailability during the listing or anticipated unavailability during the study period. This does not include individuals who refused participation (even though their household agreed to participate in the study) or could not be contacted after they were selected to participate in the study (1.4%), or individuals who were ineligible for inclusion because they were absent the previous night (10.1% of individuals in households). While absent individuals were likely more similar to those included in the missed population study, we do not know the characteristics of individuals who refused participation.

Both social desirability and recall bias may have impacted the reporting of vaccination coverage for children who did not have under-five cards or other documentation of vaccination history. Social desirability may have also affected reporting whether individuals had visited the referral-level health facilities or if they would have visited them had they been referred. It is possible that social desirability bias was more substantial in the missed population study because some individuals may have felt that they inconvenienced the data collection team due to additional efforts the team took to locate them for the missed population study. It is likely

that this would have resulted in bias away from the null in our study, which meant that the magnitude of bias estimated in our simulation is greater than the true bias resulting from missing these populations.

The findings are consistent with other published studies. For example, a study in India found that households not listed in the sampling frame were more difficult to physically access (e.g., terrain or living in residential complex), which likely resulted in unbalanced socio-demographic characteristics [35]. However, it is unclear whether these imbalances resulted in biased estimates of measles and rubella seroprevalence or other outcomes. Our study also found higher nonresponse rates in urban clusters, although data collection teams did not experience challenges in reaching households due to geographical inaccessibility. A meta-analysis of 59 studies using different sampling strategies found that nonresponse bias stemming from sampling frame exclusion resulted in an 8% difference in outcome proportions between respondent and nonrespondent groups, a lower average bias than other sources of nonresponse [36]. The authors found that the risk of bias can be reduced through high response rates when the reasons for nonresponse are highly correlated with the distribution of response variables [36]. In our study, we observed that while the rate of exclusion from the sampling frame was sizeable, and populations differed among some socio-demographic characteristics, this did not lead to significant nonresponse bias.

## Conclusions

This study found differences in demographic characteristics of populations included in a serosurvey conducted in two districts in Zambia in 2022, and those missed from the serosurvey's sampling frame. Namely, the two populations differed in household size, occupations of adults in the households, household wealth, and healthcare seeking. However, simulations showed that exclusion of households from sampling frame did not translate to meaningful differences in the outcomes of interest, resulting in less than 5% difference in estimated measles seroprevalence and vaccination coverage. In other settings where differences in socio-economic characteristics may be more strongly associated with the outcomes or with a higher percentage of missed households, this bias may impact results. Using a pre-specified threshold of missingness to trigger additional days of listing in a cluster before commencing data collection could be a way to reduce the potential for bias. Finally, this study highlights the importance of considering the study methodology and challenges encountered during implementation when interpreting the results of community-based studies including accounting for potential shortcomings in accurately establishing sampling frames. In contexts where sampling frames systematically exclude hard-to-reach populations, including mobile populations or individuals with less access to social power, we strongly recommend additional efforts to improve and enrich the sampling frame may result in biased results and potentially detrimental effects on decision-making.

## Supporting information

**S1 Checklist.**
(DOCX)

**S1 Fig. Time of data collection in original serosurvey ("original") and the missed population study ("missed population").** Gaps in data collection for the same survey indicate that during the initial visit to the cluster, the teams could not locate all households or individuals for enrollment and revisited the cluster later to complete data collection.
(PDF)

**S2 Fig. Estimates of outcomes of interest using the original sampling frame (excluding missed households) and a mixed sampling frame (including both missed households and households enrolled in the parent study).** Weighted and unweighted estimates. Weighting was done using the estimated population in each age group in each cluster in the missed population study. A. Health care seeking (actual and theoretical) at facilities of interest (Arthur Davison Children's Hospital and Choma General Hospital for children 1–4 and 5–14 years old, and Ndola Teaching Hospital and Choma General Hospital for adults 15 years and older). B. MCV2 coverage, children 1–4 years old. C. Measles seroprevalence, children 1–4 years old. (PDF)

**S1 Table. Number of households selected in each cluster in bootstrapping, by age group and district.**
(DOCX)

**S2 Table. Individual demographic characteristics of individuals enrolled in the original study and missed population study, children 1–4 years old.** The original serosurvey was carried out in April—June 2022 in Ndola and Choma districts, Zambia, using stratified multistage clustering design. The follow-up missed population study was carried out in a subset of clusters of the original survey between July—August 2022. This study was carried out in a subsample of clusters from the original survey; in each selected cluster, a sample of households not available during listing of the original serosurvey, and hence excluded from its sampling frame, were randomly selected.
(DOCX)

**S3 Table. Individual demographic characteristics of individuals enrolled in the original study and missed population study, children 5–14 years old.** The original serosurvey was carried out in April—June 2022 in Ndola and Choma districts, Zambia, using stratified multistage clustering design. The follow-up missed population study was carried out in a subset of clusters of the original survey between July—August 2022. This study was carried out in a subsample of clusters from the original survey; in each selected cluster, a sample of households not available during listing of the original serosurvey, and hence excluded from its sampling frame, were randomly selected.
(DOCX)

**S4 Table. Individual demographic characteristics of individuals enrolled in the original study and missed population study, adults 15 years and older.** The original serosurvey was carried out in April—June 2022 in Ndola and Choma districts, Zambia, using stratified multistage clustering design. The follow-up missed population study was carried out in a subset of clusters of the original survey between July—August 2022. This study was carried out in a subsample of clusters from the original survey; in each selected cluster, a sample of households not available during listing of the original serosurvey, and hence excluded from its sampling frame, were randomly selected.
(DOCX)

**S5 Table. Healthcare-seeking behavior reported by adults, stratified by living in households with or without children The original serosurvey was carried out in April—June 2022 in Ndola and Choma districts, Zambia, using stratified multi-stage clustering design.** The follow-up missed population study was carried out in a subset of clusters of the original survey between July—August 2022. This study was carried out in a subsample of clusters from the original survey; in each selected cluster, a sample of households not available during listing of the original serosurvey, and hence excluded from its sampling frame, were randomly

selected.
(DOCX)

**S6 Table. History of fever / rash symptoms in the past two weeks and diagnosis of suspected measles, children 1–4 and 5–14 years old.** The original serosurvey was carried out in April—June 2022 in Ndola and Choma districts, Zambia, using stratified multi-stage clustering design. The follow-up missed population study was carried out in a subset of clusters of the original survey between July—August 2022. This study was carried out in a subsample of clusters from the original survey; in each selected cluster, a sample of households not available during listing of the original serosurvey, and hence excluded from its sampling frame, were randomly selected.
(DOCX)

**S7 Table. Results of unweighted and weighted bootstrapping.** We used bootstrapping procedure to simulate inclusion of missed households in the sampling frame of a serosurvey carried out in Ndola and Choma districts, Zambia, in April—June 2022.
(DOCX)

**S1 Appendix. Graphical summary of study methods.**
(DOCX)

**S2 Appendix. Definitions of outcome and explanatory variables.**
(DOCX)

**S3 Appendix. Multivariable mixed-effects model.**
(DOCX)

**S4 Appendix. Selection probabilities of households in bootstrapping.**
(DOCX)

**S5 Appendix. Covariates for predictive model for measles seropositivity.**
(DOCX)

## Acknowledgments

This study would not have been possible without the dedication of our data collection teams, which put in an incredible effort to reach individuals and households that were missed in the original study. This meant working on weekends and holidays, at 5 in the morning and late at night, walking kilometers to reach some households, and evading elephant stampedes. We remain indebted to these teams. We are also grateful to the communities in Choma and Ndola Districts that welcomed our investigators into their homes, and survey participants that shared their time and experiences with us. We are highly appreciative of their trust in our teams.

## Author Contributions

**Conceptualization:** Natalya Kostandova, Francis Dien Mwansa, Bryan Lau, William J. Moss, Amy Wesolowski.

**Data curation:** Natalya Kostandova, Christine Prosperi, Chola Nakazwe, Harriet Namukoko, Bertha Nachinga.

**Formal analysis:** Natalya Kostandova.

**Funding acquisition:** William J. Moss, Amy Wesolowski.

**Investigation:** Chola Nakazwe, Harriet Namukoko, Bertha Nachinga, Innocent Chilumba, Kalumbu H. Matakala, Gloria Musukwa, Mutinta Hamahuwa, Webster Mufwambi, Japhet Matoba, Kenny Situtu, Irene Mutale.

**Methodology:** Natalya Kostandova, Simon Mutembo, Christine Prosperi, Chola Nakazwe, Harriet Namukoko, Bertha Nachinga, Alvira Z. Hasan, Shaun A. Truelove, Amy K. Winter, Andrea C. Carcelen, Bryan Lau, Amy Wesolowski.

**Project administration:** Simon Mutembo, Gershom Chongwe, Edgar Simulundu, Phillimon Ndubani, Bryan Lau, William J. Moss, Amy Wesolowski.

**Resources:** Simon Mutembo, Christine Prosperi, Chola Nakazwe, Harriet Namukoko, Bertha Nachinga, Gershom Chongwe, Innocent Chilumba, Kalumbu H. Matakala, Gloria Musukwa, Mutinta Hamahuwa, Irene Mutale, Edgar Simulundu, Phillimon Ndubani, Alvira Z. Hasan, Andrea C. Carcelen, Bryan Lau.

**Software:** Natalya Kostandova, Christine Prosperi, Chola Nakazwe, Harriet Namukoko, Bertha Nachinga, Innocent Chilumba, Webster Mufwambi, Japhet Matoba, Alvira Z. Hasan, Shaun A. Truelove.

**Supervision:** Natalya Kostandova, Simon Mutembo, Christine Prosperi, Francis Dien Mwansa, Chola Nakazwe, Harriet Namukoko, Bertha Nachinga, Gershom Chongwe, Innocent Chilumba, Kalumbu H. Matakala, Gloria Musukwa, Mutinta Hamahuwa, Webster Mufwambi, Japhet Matoba, Kenny Situtu, Irene Mutale, Edgar Simulundu, Phillimon Ndubani, Alvira Z. Hasan, Shaun A. Truelove, Bryan Lau, William J. Moss, Amy Wesolowski.

**Validation:** Chola Nakazwe, Harriet Namukoko, Bertha Nachinga.

**Visualization:** Natalya Kostandova, Alex C. Kong.

**Writing – original draft:** Natalya Kostandova.

**Writing – review & editing:** Natalya Kostandova, Simon Mutembo, Christine Prosperi, Francis Dien Mwansa, Alex C. Kong, Phillimon Ndubani, Amy K. Winter, Andrea C. Carcelen, Bryan Lau, William J. Moss, Amy Wesolowski.

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
