## [Decision Letter · Decision Letter 0]

19 Oct 2023

PGPH-D-23-01791

Who is missed in a community-based survey: Assessment and implications of biases due to missed populations in Zambia community-based serosurvey

Dear Kostadonva,

Thank you for submitting your manuscript to PLOS Global Public Health. After careful consideration, we feel that it has merit but does not fully meet PLOS Global Public Health’s publication criteria as it currently stands. Therefore, we invite you to submit a revised version of the manuscript that addresses the points raised during the review process.

We look forward to receiving your revised manuscript.

Kind regards,

Collins Otieno Asweto, PhD

Academic Editor

Journal Requirements:

Reviewers' comments:

Reviewer's Responses to Questions

**Comments to the Author**

1. Does this manuscript meet PLOS Global Public Health’s publication criteria? Is the manuscript technically sound, and do the data support the conclusions? The manuscript must describe methodologically and ethically rigorous research with conclusions that are appropriately drawn based on the data presented.

Reviewer #1: No

Reviewer #2: Yes

Reviewer #3: Yes

2. Has the statistical analysis been performed appropriately and rigorously?

Reviewer #1: No

Reviewer #2: Yes

Reviewer #3: Yes

3. Have the authors made all data underlying the findings in their manuscript fully available (please refer to the Data Availability Statement at the start of the manuscript PDF file)?

Reviewer #1: No

Reviewer #2: Yes

Reviewer #3: Yes

4. Is the manuscript presented in an intelligible fashion and written in standard English?

Reviewer #1: Yes

Reviewer #2: Yes

Reviewer #3: Yes

5. Review Comments to the Author

Reviewer #1: The authors of this manuscript attempt to assess the extent to which sampling bias could affect the estimates of prevalence of measles immunity in community samples. However, their own study appear to suffer from the very own criticisms of sampling bias that they claim to address. The methodological limitations of this study are obvious from the onset. First, they characterize the ZAMPHIA sample as a "convenience sample". That is an inaccurate statement and raise serious concerns over the authors understanding of the complex sampling procedures employed in ZAMPHIA. The sample of ZAMPHIA was generated using a "stratified multistage probability sample design, with strata defined by the 10 provinces of Zambia, first-stage sampling units defined by enumeration areas (EAs) within strata, second-stage sampling units defined by households within EAs, and finally eligible persons 0-59 years of age within households."(https://phia-data.icap.columbia.edu/storage/Country/26-10-2021-16-38-29-61782f050999c.pdf). To call the ZAMPHIA sample a "convenience sample" is a gross misunderstanding of sampling procedures. Second, the study makes no mention of the Zambia Demographic and Health Survey (DHS) 2018 (https://dhsprogram.com/pubs/pdf/FR361/FR361.pdf), another state-of-the-science community-based survey employing complex sampling strategies. Third, there is not a single mention of the World Health Organization Expanded Program for Immunization (WHO-EPI) random walk sampling method, nor the compact segment sampling method (https://academic.oup.com/ije/article/33/3/469/716634?login=false). The omission of the Zambia DHS, and a review of its sampling methods, as well as an omission of the WHO-EPI and compact segment sampling methods show that the authors did not even review the literature on sampling for epidemiological studies appropriately. If anything, the sample used by this study is much more a convenience sample that the one obtained by ZAMPHIA, the DHS, or any sampling using the WHO-EPI or the compact segment methods. The authors use a sample that only included two districts out of 116 districts in Zambia and provide no explanation as to the sampling procedures used to obtain the sample included in this study. Consequently, the findings of this study can only be expect to refer to the inadequacies of the original, poorly designed, study that they based their research on.

Reviewer #2: The main question addressed by this research is very relevant and interesting. As part of efforts to quantify measles immunity gaps in Zambia, a community-based serological survey was conducted in two districts. This Community-based serological studies is well written with clear context, easy to read with the presentations and conclusions drawn consistent with arguments and evidence presented by the authors. The figures and tables aid better understanding of the work done. This research is comprehensive especially in terms of the presented content, methodology, ethical considerations and originality. The abstract is however too long and should be shortened. The authors may wish to throw more light on rational for choosing measles and why this serosurvey was conducted in Choma and Ndola Districts ( two districts) in Zambia.

Reviewer #3: This is an excellent manuscript which is technically sound. Consistency has been maintained throughout its aim, research questions, methods, results, discussion and conclusions. The methodology has been clearly described and ethical permissions were sought from appropriate authorities. Data are presented using graphs and tables and conclusions are drawn base of the data.

Appropriate statistical tools were used. Though the tables with statistical analysis may not be useful for lay readers. It may be worth discussing with a statisticians to keep the tables with high level data which may be more understandable to wider readers. The original tables can be included in the supplementary documents.

There is an explicit mention of the data sharing restrictions in the 'Data Availability' section and how the corresponding author can coordinate a request to the Ministry of Health of Zambia to access those data.

The language used in the manuscript is of high standard in a clear, correct, and unambiguous fashion. Any typographical or grammatical errors should be corrected at revision, so please note any specific errors here. Some minor typos are present that need to be corrected by spell-check tools. Some of these typos are as follows:

Line 33: magnitude of (missing)

34: made not make; in not on

37: distributions

41: reducing

56: The (missing) key

71: have not has

77: execution (no comma) and the (missing)

104: age (no hyphen) range

122: household (no comma)

125: two days

126: information (no comma) ...... who (not that) knew

141: to (should be removed)

164: were (not was)

170: population

260: studies

316: hospitals

458: a (missing) sub-national

6. PLOS authors have the option to publish the peer review history of their article (what does this mean?). If published, this will include your full peer review and any attached files.

**Do you want your identity to be public for this peer review?** For information about this choice, including consent withdrawal, please see our Privacy Policy.

Reviewer #1: No

Reviewer #2: **Yes: **PRISCILIA UHUANMWEN IMADE

Reviewer #3: No

---

## [Decision Letter · Decision Letter 1]

18 Jan 2024

PGPH-D-23-01791R1

Who is missed in a community-based survey: Assessment and implications of biases due to missed populations in Zambia community-based serosurvey

Dear Kostandova,

Thank you for submitting your manuscript to PLOS Global Public Health. After careful consideration, we feel that it has merit but does not fully meet PLOS Global Public Health’s publication criteria as it currently stands. Therefore, we invite you to submit a revised version of the manuscript that addresses the points raised during the review process.

We look forward to receiving your revised manuscript.

Kind regards,

Collins Otieno Asweto, PhD

Academic Editor

Journal Requirements:

Reviewers' comments:

Reviewer's Responses to Questions

**Comments to the Author**

1. If the authors have adequately addressed your comments raised in a previous round of review and you feel that this manuscript is now acceptable for publication, you may indicate that here to bypass the “Comments to the Author” section, enter your conflict of interest statement in the “Confidential to Editor” section, and submit your "Accept" recommendation.

Reviewer #4: (No Response)

Reviewer #5: (No Response)

2. Does this manuscript meet PLOS Global Public Health’s publication criteria? Is the manuscript technically sound, and do the data support the conclusions? The manuscript must describe methodologically and ethically rigorous research with conclusions that are appropriately drawn based on the data presented.

Reviewer #4: (No Response)

Reviewer #5: Yes

3. Has the statistical analysis been performed appropriately and rigorously?

Reviewer #4: (No Response)

Reviewer #5: Yes

4. Have the authors made all data underlying the findings in their manuscript fully available (please refer to the Data Availability Statement at the start of the manuscript PDF file)?

Reviewer #4: (No Response)

Reviewer #5: Yes

5. Is the manuscript presented in an intelligible fashion and written in standard English?

Reviewer #4: (No Response)

Reviewer #5: Yes

6. Review Comments to the Author

Reviewer #4: Please address review comments

Reviewer #5: 1. The number of households selected randomly in each cluster was 40 (Within each selected cluster, all missing households were eligible if the number of missing households was less than 40; otherwise, 40 households were randomly selected): How was this number determined? Why not 30, 45, 50, …? A brief explanation should be included in the final version).

2. The missed population study collection was conducted by a subset of data collection personnel involved in the original study: What was the advantage of using data collection staff involved in the original study? Minimization of the risk of interviewer bias? Why not a new data collection staff or a mixed data collection team? A brief explanation of your choice precisely on mitigating/controlling the risk of interviewer bias is desirable.

3. Regarding individuals agreed to participate to the survey, data collection teams conducted interviews using a shortened data collection tool from the original study: Was the data collection tool used in the original study: standard? adapted or developed? In all cases, how the content of the shortened data collection tool was assessed acceptable? A brief description should be provided in the final version of the document.

4. The investigation was carried out according to methods recognized as being rigorous based on proven mathematical theories: What softwares were used for data processing? The same as for the initial study? Please specify in the Method section.

Recommendation: For each of the four questions above, please provide a brief explanation/description/justification in the final version of the paper (Method and/or Discussion section).

7. PLOS authors have the option to publish the peer review history of their article (what does this mean?). If published, this will include your full peer review and any attached files.

**Do you want your identity to be public for this peer review?** For information about this choice, including consent withdrawal, please see our Privacy Policy.

Reviewer #4: No

Reviewer #5: **Yes: **Lazare M'BOUNGOU

---

## [Decision Letter · Decision Letter 2]

12 Mar 2024

Who is missed in a community-based survey: Assessment and implications of biases due to incomplete sampling frame in a community-based serosurvey, Choma and Ndola Districts, Zambia, 2022

PGPH-D-23-01791R2

Dear Natalya,

We are pleased to inform you that your manuscript 'Who is missed in a community-based survey: Assessment and implications of biases due to incomplete sampling frame in a community-based serosurvey, Choma and Ndola Districts, Zambia, 2022' has been provisionally accepted for publication in PLOS Global Public Health.

Best regards,

Collins Otieno Asweto, PhD

Academic Editor

Reviewer Comments (if any, and for reference):

Reviewer's Responses to Questions

**Comments to the Author**

1. If the authors have adequately addressed your comments raised in a previous round of review and you feel that this manuscript is now acceptable for publication, you may indicate that here to bypass the “Comments to the Author” section, enter your conflict of interest statement in the “Confidential to Editor” section, and submit your "Accept" recommendation.

Reviewer #2: All comments have been addressed

Reviewer #6: All comments have been addressed

2. Does this manuscript meet PLOS Global Public Health’s publication criteria? Is the manuscript technically sound, and do the data support the conclusions? The manuscript must describe methodologically and ethically rigorous research with conclusions that are appropriately drawn based on the data presented.

Reviewer #2: Yes

Reviewer #6: Yes

3. Has the statistical analysis been performed appropriately and rigorously?

Reviewer #2: Yes

Reviewer #6: Yes

4. Have the authors made all data underlying the findings in their manuscript fully available (please refer to the Data Availability Statement at the start of the manuscript PDF file)?

Reviewer #2: Yes

Reviewer #6: Yes

5. Is the manuscript presented in an intelligible fashion and written in standard English?

Reviewer #2: Yes

Reviewer #6: Yes

6. Review Comments to the Author

Reviewer #2: The authors have done well by addressing all the concerns raised.

Reviewer #6: Methodology Section:

1. Break down paragraphs for clarity, use visual aids, and provide more details on missed households and sampling.

2. Elaborate on data collection tools, explain variables, and clarify mixed-effects model rationale.

3. Proofread for grammatical errors to improve readability.

Operational Definition Section:

1. Clearly define "missed households" and criteria for exclusion.

2. Explain the rationale behind chosen power parameters for the missed population survey.

Results:

The results are well-organized, use appropriate statistical tests, and provide context. The study's strengths and weaknesses are transparently presented.

Discussion:

The discussion is structured, insightful, and interprets biases and limitations effectively. It adds valuable insights to research discourse, acknowledges biases, and reflects on social desirability. Overall, it provides a comprehensive reflection on research outcomes.

7. PLOS authors have the option to publish the peer review history of their article (what does this mean?). If published, this will include your full peer review and any attached files.

**Do you want your identity to be public for this peer review?** For information about this choice, including consent withdrawal, please see our Privacy Policy.

Reviewer #2: **Yes: **PRISCILIA UHUANMWEN IMADE

Reviewer #6: **Yes: **Giri sarashwati Giri
